# Recovery from Hypersaline-Stress-Induced Immunity Damage and Intestinal-Microbiota Changes through Dietary β-glucan Supplementation in Nile tilapia (*Oreochromis niloticus*)

**DOI:** 10.3390/ani10122243

**Published:** 2020-11-30

**Authors:** Chang Xu, Yantong Suo, Xiaodan Wang, Jian G Qin, Liqiao Chen, Erchao Li

**Affiliations:** 1Key Laboratory of Tropical Hydrobiology and Biotechnology of Hainan Province, Hainan Aquaculture Breeding Engineering Research Center, College of Marine Sciences, Hainan University, Haikou 570228, China; cxu@hainanu.edu.cn; 2School of Life Sciences, East China Normal University, Shanghai 200241, China; suoyantong@163.com (Y.S.); xdwang@bio.ecnu.edu.cn (X.W.); lqchen@bio.ecnu.edu.cn (L.C.); 3School of Biological Sciences, Flinders University, Adelaide, SA 5001, Australia; jian.qin@flinders.edu.au

**Keywords:** Nile tilapia, immune function, hematology, intestinal microbiota

## Abstract

**Simple Summary:**

Long-term hypersaline stress can induce coagulation disorders and splenomegaly and down-regulate the complement pathway in tilapia, which can increase risk in healthy breeding. As a prebiotic, β-glucan dietary supplementation can significantly reduce enlarged spleen resulting from hypersaline stress. The hematological aspects of the red blood cell count, hematocrit, red cell distribution width, platelet count, and plateletcrit were also decreased by supplementation with dietary β-glucan. In the spleen and intestine, β-glucan intake significantly decreased the high expression of immune-related genes due to hypersaline stress resulting from β-glucan intake in tilapia. β-glucan supplementation also significantly increased the abundance of beneficial microbiota such as *Lactobacillus*, *Phycicoccus,* and *Rikenellaceae* in the intestine. In summary, β-glucan intake can relieve tissue damage and optimize the intestinal microbiota of tilapia in brackish water and improve fish health.

**Abstract:**

Long-term exposure to hyperosmotic environments can induce severe immune damage and increase risk in tilapia breeding. As an effective immunoregulator, β-glucan has attracted extensive attention in nutritional research and given rise to high expectations of improving health status and alleviating organismal damage in tilapia, *Oreochromis niloticus,* in brackish water. In this study, an 8-week cultivation experiment was conducted on tilapia fed a basal diet or diets with β-glucan supplementation in freshwater (control) and brackish water. Growth performance, hematological aspects, immune cytokine expression, and the intestinal microbiota of tilapia were analyzed. The results indicated that supplementation with β-glucan significantly reduced the enlarged spleen of tilapia resulting from hypersaline stress. Tilapia fed β-glucan showed significantly-greater decreases in the red blood cell count, hematocrit, red cell distribution width, platelet count, and plateletcrit than those fed the basal diet. β-glucan significantly decreased the high expression of immune-related genes in the spleen induced by hyperosmotic stress. In the intestine, the high migration inhibitory factor-2 (*MIF-2*) and *IL-1β* gene expression induced by hypersaline stress was significantly reduced. β-glucan supplementation also significantly increased the abundance of beneficial microbiota such as *Lactobacillus*, *Phycicoccus,* and *Rikenellaceae*. Therefore, dietary β-glucan supplementation can significantly reduce spleen enlargement and improve immune function in tilapia in brackish water. β-glucan intake can also optimize the intestinal microbiota of tilapia in brackish water and improve fish health.

## 1. Introduction

As a biologically-active immunomodulator, β-glucan is a ubiquitous component of fungi, yeast, oats, and seaweed [1]. β-glucan can be recognized by the innate immune system, which plays an essential role in host defense [2]. In pharmaceutics, β-glucan shows potential therapeutic value for immune improvement, along with anti-inflammatory and anti-cancer effects [3,4]. As a dietary additive, the positive effect of β-glucan is also found in various aquatic animals under stress conditions. Resistance capacity against bacterial infection and immune function can be enhanced by β-glucan addition in *Oreochromis niloticus* [5,6]. β-glucan supplementation also can improve growth and immune function in zebrafish [7], *Pagrus major* [8], *Ictalurus punctatus* [9], and *Pseudosciaena crocea* [10]. In *Litopenaeus vannamei*, dietary β-glucan can significantly improve growth performance and induce higher respiratory burst and higher richness of probiotics in the intestine versus basal diets at low salinity [11,12].

In mammals, β-glucans have a significant impact on the intestinal microbiota, and in turn on organismal health [13]. In addition to its effective immunomodulatory properties, β-glucan is also a typical fiber digested by intestinal enzymes to produce short-chain fatty acids (SCFAs) [14]. SCFAs are beneficial to reduce the pH and are thus beneficial for the competitive exclusion of pathogens in the intestine [15]. Intestinal bacterial metabolism in aquatic animals shows important functions in health regulation and nutrient absorption [16]. Dietary β-glucan supplementation can improve intestinal microbiota communities in *Cyprinus carpio* [17], *Scophthalmus maximus* [18], *Rutilus Frisii kutum* [19], and *Cyprinus carpio* L [20]. β-glucans can also increase the dominance of the intestinal microbiota richness in *L. vannamei* under the conditions of ammonia and low-salinity stress [12,21].

For aquatic animals, salinity is a key ecological index that directly influences growth, physiological status, immune function, and nutritional value [22]. In our previous research, chronic hyperosmotic stress induced an intumescent spleen and enlarged macrophage centers in head kidney and suppressed immune function in *O. niloticus* [23]. As the two largest lymphoid and immunocompetent organs, the spleen and head kidney are sensitive to environmental changes [24,25]. In *O. mossambicus*, acute salinity stress can increase phagocytosis, respiratory burst activity, and humoral immune reactions in the spleen and head kidney [26]. Chronic salinity stress can also increase susceptibility to streptococcus infection in *O. mossambicus* [27]. After exposure to endosulfan at a level equivalent to ½ LC50 for 96 h, a lower spleen cell viability and relative spleen weight were found in *O. niloticus* [28]. Similarly, exposure to cadmium chloride (20.93 mg/L) for 120 h can cause significant changes in melano-macrophage centers (MMCs) and free macrophages in the spleen and head kidney of *O. mossambicus* [29]. The blood parameters of teleost fish are also sensitive to salinity stress, as evidenced in *Huso* [30], *Rachycentron canadum* [31], *Notopterus notopterus* [32], and *Acipenser brevirostrum* [33].

Nile tilapia (*Oreochromis niloticus*) is the most economically-important and extensively-cultured aquatic species worldwide. Although tilapia exhibit strong adaptation to environmental salinity, hypersaline stress can still damage immune function and increase the risks of disease and death. Previous research has indicated that environmental salinity can significantly alter the gut microbiota and induce impaired immune function in *O. niloticus* [23,34]. Therefore, it is necessary to find an effective method to mitigate or eliminate the negative impacts in tilapia induced by environmental hyperosmotic stress. In this study, the effects of β-glucan supplementation in the diet on growth performance, hematology, histology, immunity gene expression, and intestinal microbiota communities were analyzed to evaluate the comprehensive health status response in Nile tilapia under hypersaline environments. This study will provide new insights to explore the role of dietary nutrient manipulation in fish to cope with environmental stress.

## 2. Materials and Methods

### 2.1. Ethical Approval

The animal ethics protocol was approved by the East China Normal University Experimental Animal Ethics Committee (No. f20190201).

### 2.2. Experimental Diets

Isonitrogenous and isoenergetic experimental diets were formulated with three concentrations of β-glucan (0, 0.2 and 0.4%). Raw materials were crushed with a pulverizer and sieved through a 60-μm mesh and then mixed evenly. Each mixture was dissolved by adding deionized water and then wet-extruded into 2.5-mm-diameter pellets using a double-helix plodder (F-26, SCUT industrial factory, Guangdong, China). The scattered pellets were air-dried at room temperature until <10% moisture content was attained. Pellets were graded to various sizes by 12- and 8-mesh sieves and stored at −20 °C until use. Ingredient and proximate compositions of the three experimental diets are given in Table 1.

### 2.3. Experimental Fish and Management Procedure

The juvenile Nile tilapia (all males) were obtained from a local farm in Hainan, China. All fish were acclimated in four tanks for one week with commercial diets. During acclimation, the salinity in the three tanks was gradually increased to 16 practical salinity units (psu) at a daily rate of 4 psu by adding sea salt. After the salinity reached the target value, fish with an initial weight of 1.28 ± 0.03 g were randomly assigned to 16 tanks (750 L) at a density of 35 fish per tank. There were four experimental treatments (freshwater-0% β-glucan, 16 psu-0% β-glucan, 16 psu-0.2% β-glucan, and 16 psu-0.4% β-glucan) with four replicates each. During the 53-day experiment, fish were fed to satiation twice daily (0800 and 1400 h). Two tanks (freshwater and brackish water) were used for water storage before entering the experimental system. The daily water exchange rate was 50% of the water volume. Animals were illuminated by natural light with a specific photoperiod. Water quality parameters were checked every three days. The concentration of dissolved oxygen was 6.5–7.5 mg/L, pH averaged 7.50 ± 0.30, ammonia-N was < 0.05, and water temperature averaged 30 ± 2 °C.

### 2.4. Sample Collection

At the end of the cultivation, all fish were fasted for 24 h. Subsequently, the fish were anesthetized in 30 ppm MS-222 before being counted and weighed to determine survival and weight gain [35,36]. The blood of each fish was individually sampled by caudal sinus puncture with plastic sterile syringes. Blood samples were transferred to lithium heparin tubes for hematological determination. The spleen, head kidney, and mid-intestine of 15 fish in each tank were dissected quickly and then stored in liquid nitrogen to analyze immune gene mRNA and intestinal microflora. The spleens of five fish in each tank were dissected and weighed to determine relative spleen weight. Survival, weight gain, and relative spleen weight were calculated as follows:

Survival (%) = (final fish number/initial fish number) × 100

Weight gain (WG, %) = (final weight (g) − initial weight (g))/initial weight (g) × 100

Relative spleen weight (RSW, %) = (wet spleen weight)/(wet body weight) × 100

### 2.5. Hematological Assay

The red blood cell count (RBC, 1,012/L), hematocrit (HCT, %), red cell distribution width (RDW, %), white blood cell count (WBC, 109/L), platelet count (PLT, 109/L) and plateletcrit (PCT, %) were investigated by an automated hematological analyzer (BC-2800vet, Shenzhen, Mindray Bio-Medical Electronics, Shenzhen, China).

### 2.6. Immune-Related Gene Expression

Total RNA in the spleen, head kidney, and intestine was extracted with the TRIzol reagent (RN0101, Aidlab, Beijing, China). RNA quality and quantity were analyzed using a NanoDrop 2000 spectrophotometer (Thermo, Wilmington, NC, USA). After the quantity was analyzed, 1 μg total RNA was reverse transcribed using the PrimeScripTM RT Master Mix (RR047A, Takara, Japan). The system program was 42 °C for 2 min to remove genomic DNA followed by 37 °C for 15 min and 85 °C for 5 s to complete reverse transcription. Six replicates per treatment were run for each gene with elongation factor-1α (*EF-1α*) as the internal control. Primers of interleukin-1β (*IL-1β*), migration-inhibitory factor-2 (*MIF-2*), transforming growth factor-β1 (*TGF- β1*), and tumor necrosis factor-α (*TNF-α*) were designed and validated by Primer Premier 6.0 according to National Coalition Building Institute (NCBI) (Table 2).

Real-time PCR (RT-PCR) was performed in a volume of 20 μL containing 10 μL 2×Ultra SYBR mixture (CW0957, Kangwei, China), 0.5 μL of 10 mM primers, 2.5 μL of diluted first-standard cDNA template and 6.5 μL of H_2_O. The program cycling conditions were denaturation at 95 °C for 30 s, followed by 40 cycles of 94 °C for 15 s, 58 °C for 20 s, 72 °C for 20 s, and a 0.5 °C per 5 s increment from 60 °C to 95 °C. RT-PCR was analyzed in the CFX96 Real-Time PCR system (Bio-Rad, Richmond, CA, USA). The cycle time (Ct) values of different experimental treatments in hypersaline water were compared to their corresponding internal control and then converted to fold change by comparison with the freshwater treatment through the quantified method of 2^−ΔΔct^. CFX Manager™ Software (version 1.0) was used to study data visualization and relative quantification analysis. 

### 2.7. Intestinal Microbiota Analysis in Nile tilapia

Intestinal contents were used for bacterial composition analysis in Nile tilapia under four experimental treatments. The total microbiota DNA was isolated with an E.Z.N.A.TM soil DNA kit according to the manufacturer’s instructions. DNA quality was assessed by PCR amplification of the bacterial 16S rRNA genes, and the quantity was measured using a NanoDrop spectrophotometer (Thermo, Wilmington, DE, USA). The V3–V4 region of the bacterial 16S ribosomal RNA genes was amplified by PCR using the primers 338F (ACTCCTACGGGAGGCAGCA) and 806R (GGACTACHVGGGTWTCTAAT). The PCR reactions were performed in a 25 μL mixture containing 5 μL of 5× reaction buffer, 5 μL of 5× GC buffer, 2 μL of 2.5 mM dNTPs, 1 μL of forward primer (10 μM), 1 μL of reverse primer (10 μM), 2 μL of DNA template, 8.75 μL of H_2_O, and 0.25 μL of Q5 high-fidelity DNA polymerase (M0419S, New England BioLabs, Beijing, China). The PCR reaction conditions were as follows: 98 °C for 2 min, followed by 26 cycles of denaturation at 98 °C for 15 s, annealing at 55 °C for 30 s, extension at 72 °C for 30 s, and extension at 72 °C for 5 min. The purified PCR products were subjected to the Illumina MiSeq PE300 platform (Shanghai Personal Biotechnology Co., Ltd., Shanghai, China), generating paired-end reads. Low-quality sequences with a length below 150 bp, and mononucleotide repeats over 8 bp, average Phred scores <20 and with ambiguous bases were removed. The sequences obtained in the present study can be downloaded from Sequence ReadArchive (SRA) with the accession number PRJNA433775.

Operational taxonomic units (OTUs) were clustered according to the similarity cutoff of 97% using UPARSE (version 7.1 http://drive5.com/uparse/) (Edgar, 2013). The most abundant sequence in the OTUwas selected as the representative sequence, followed by taxonomic assignment in the Greengenes database (release 13.8) using the confidence threshold of 70%. QIIME calculated alpha diversity indices. Partial least squares discriminant analysis (PLS-DA) was used to test classification model influence by R software. A Venn diagram was constructed to identify the same and unique OTUs. Intestinal microbiota interspecies interaction among dominant classes with abundance in the top 50 of four treatments was calculated using Mothur, constructing the interspecies interaction network for the dominant class with Rho > 0.6 and *p* < 0.01. The network properties were calculated and visualized with Gephi.

### 2.8. Statistical Analysis

Statistical analysis was performed using SPSS statistics 20 (IBM, Armonk, NY, USA). All data were mean ± standard error (mean ± S.E.). Significant differences in relative spleen weight and intestinal microbiota richness and diversity were analyzed by one-way analysis of variance (ANOVA) in tilapia among all treatments. Data on hematology and mRNA expression in tilapia in brackish water were analyzed by ANOVA. Student’s t-test was used to analyze significant differences in hematological data and mRNA expression in tilapia between freshwater and brackish water. A total of 37 OTUs were selected for heatmap analysis using Student’s t-test with subsequent Bonferroni correction. A single asterisk “*” represents significant differences (*p* < 0.05), and double asterisks “**” represent extremely-significant differences (*p* < 0.01).

## 3. Results

### 3.1. Survival, Growth Performance, and Relative Spleen Weight

There were no significant differences in weight gain (WG) or survival among all treatments. Tilapia in hypersaline water showed a significantly-higher relative spleen weight (RSW) than those in freshwater (*p* < 0.05). The supplementation of β-glucan significantly reduced RSW in tilapia in brackish water (*p* < 0.05) (Table 3).

### 3.2. Hematological Parameters

Brackish water significantly increased RBC, HCT, RDW, PLT, and PCT in the blood of tilapia (*p* < 0.05). Tilapia fed diets with β-glucan showed significantly-lower RBC, HCT, RDW, PLT, and PCT than tilapia in brackish water fed the basal diet (*p* < 0.05) (Figure 1).

### 3.3. Gene Expression of Immune-Inflammatory Cytokines

Tilapia in brackish water exhibited significantly-higher gene expression of *TNF-α*, *TGF-β*, and *IL-1β* in the spleen than in freshwater (*p* < 0.05). Supplementation with β-glucan significantly reduced the gene expression of these immune genes (*p* < 0.05) (Figure 2A,B,D). The supplementation of β-glucan significantly decreased gene expression of *TNF-α* in a dose-dependent manner. In the intestine, gene expression of *MIF-2* and *IL-1β* was significantly increased in a brackish water environment (*p* < 0.05). These increments can be alleviated in tilapia when fed diets with β-glucan (Figure 3C,D). *TGF-β* expression in the intestine of tilapia fed the basal diet and the 0.2% β-glucan diets in brackish water was significantly higher than that in freshwater (*p* < 0.05) (Figure 3B). However, the gene expression of *TNF-α* was significantly higher in tilapia fed diets with β-glucan in brackish water than in freshwater (*p* < 0.05) (Figure 3A). In the head kidney, the *TNF-α* expression of tilapia in brackish water was significantly higher than that in tilapia in freshwater (*p* < 0.05). Tilapia in brackish water fed diets with β-glucan showed significantly-higher gene expression of *IL-1β* than in freshwater (*p* < 0.05) (Figure 4).

### 3.4. Differences in Bacterial Community Composition, Diversity, and Structure

A total of 351,714 high-quality sequences were obtained from the intestinal microbiota with an average of 87,929 sequences per sample. Microbiota richness of Chao 1 and Ace ranged from 564.00 to 939.67 and from 564.11 to 939.67, respectively. Diversity of Shannon and Simpson ranged from 5.46 to 7.02 and from 0.93 to 0.97, respectively (Table 4). Supplementation with 0.4% β-glucan significantly reduced Chao 1, ACE Shannon, and Simpson indexes in the intestine as compared with those in tilapia fed the basal diet in brackish water (*p* < 0.05). Although 0.2% β-glucan addition in diets also reduced intestinal microbiota richness and diversity as compared with absence of supplementation in the basal diet in brackish water, there was no significant difference between these two treatments.

The top 20 taxa of relative abundance were differentiated by letters in the graph (outer layers to inner layers in the order from phylum to genus) (Figure 5A). Proteobacteria showed the highest abundance, followed by Actinobacteria, Firmicutes and Bacteroidetes, which were the dominant bacteria in the intestine across all treatments.

In the Venn diagram, there were 377 OTUs shared in tilapia among four treatments. Compared with the tilapia intestinal microbiota in freshwater (648 OTUs), there were 933 in 16 psu-0% β-glucan, 473 in 16 psu-0.2% β-glucan, and 264 unique OTUs in 16 psu-0.4% β-glucan (Figure 5B). Supplementation with dietary β-glucan reduced the quantity of unique OTUs in tilapia as compared with the diet without β-glucan supplementation in freshwater and brackish water.

As shown by PLS-DA (partial least squares discriminant analysis) (Figure 5C), the distributions of bacterial communities in tilapia in freshwater and brackish water were distinguishable and showed a far coordinate distance. Supplementation with β-glucan reduced the coordinate distance, making it closer to that of the freshwater control, especially in the case of 0.4% β-glucan treatment. There was an obvious overlap of the bacteria community distributions of tilapia in brackish water fed diets with 0.2% and 0.4% β-glucan.

The heatmap analysis included 37 OTUs (Figure 6). *Moraxella* and *Leuzea* decreased significantly in tilapia in brackish water as compared with those in freshwater (*p* < 0.05). The abundances of *Thermoactinomycetacae*, *Parachlamydiaceae*, *Caldilineaceae*, *Solirubrobacteraceae*, *Microbacteriaceae,* and *Phyllobacteriaceae* were significantly lower in tilapia fed 0.2% β-glucan (*p* < 0.05) in brackish water than in freshwater. Tilapia fed 0.2 and 0.4% β-glucan exhibited significantly higher *Lactobacillus*, *Phycicoccus* and *Collinsella* than those in freshwater (*p* < 0.05). Meanwhile, *Aeromonadaceae* and *Rikenellaceae* showed a significantly-higher abundance in tilapia fed 0.2% β-glucan than those in freshwater (*p* < 0.01). *Finegoldia* and *Sediminibacterium* exhibited significantly higher abundance in tilapia fed 0.4% β-glucan than in freshwater (*p* < 0.05). Compared to tilapia fed the basal diet in brackish water, 0.4% β-glucan supplementation significantly increased *Nitrospira* and *Streptomyces* richness in tilapia (*p* < 0.05).

### 3.5. Intestinal Microbiota Interspecies Interaction

The interspecies interaction network was analyzed to evaluate the influence of environmental salinity and dietary β-glucan supplementation on interspecies interactions among the intestinal microbiota. Compared to the freshwater control, the network in *O. niloticus* in brackish water was more complex and better connected. The networks were more complex when β-glucan supplementation was provided; a dose-dependent effect was exhibited (Figure 7). The topological properties also indicated that the average path in tilapia fed β-glucan was higher than that of tilapia fed the basal diet (Table 5). There were increases in edges, average degree, graph density, centralization, and heterogeneity in the intestinal microbiota when β-glucan was supplemented in diets. β-glucan supplementation at 0.4% significantly improved the negative association ratio with respect to the results under 0.2% β-glucan supplementation.

## 4. Discussion

As an immunostimulant, β-glucan is considered an effective reagent for enhancing the immune status of aquatic animals in aquaculture [37]. β-glucan supplementation in diets can significantly improve growth performance in various fish, such as *Cyprinus carpio koi* [38], *Oncorhynchus mykiss* [39], *O. niloticus* [40], and *Apostichopus japonicas* [41]. However, in the present study, weight gain was not improved by dietary β-glucan in tilapia in brackish water. In *O. mossambicus*, only 1% β-glucan in the diet significantly increased weight gain after an 8-week cultivation [42]. Similar results were also found in *Dicentrachus labrax* [43], *O. nilotics* [44], and *Carassius auratus* by adding β-glucan [45]. Therefore, the supplementary dose of β-glucan in the diet has a key influence on growth performance, especially in a stressful environment.

Although no significant improvement in growth performance was observed in tilapia, the tissue damage was significantly alleviated by dietary β-glucan supplementation in this study. The enlargement of the spleen due to long-term hypersaline stress can be significantly improved by β-glucan supplementation in the diet regardless of concentrations. β-glucan (0.1%) in the diet also can decrease RSW in *O. niloticus* under diazinon stress [46]. Cytologically, the spleen is the largest lymphoid organ and an immunocompetent organ in teleost that is rich in melano-macrophages for phagocytosis in the immune response [47]. The normal volume of the spleen is vital to physiological function in fish. Previous research indicated that fungal polysaccharides can activate the innate immune system by binding to receptors on macrophages to stimulate the phagocytosis index in vertebrates [48]. Therefore, the spleen can be recovered to a healthy status in tilapia after an 8-week β-glucan intake under a hyperosmotic environment.

In addition to its immune function role, the spleen also serves a hematopoietic function equivalent to erythropoiesis in the bone marrow until adulthood [47]. The hematological results indicated that the RBC count was significantly higher in fish fed the basal diet in brackish water than in freshwater. Long-term hyperosmotic stress can inhibit coagulation cascades in the spleen and result in an enlarged spleen in paraffin sections from *O. niloticus* [23]. β-glucan supplementation could significantly decrease HCT, RDW, PLT, and PCT in *O. niloticus* in brackish water. However, this result is at odds with previous studies where β-glucan increased WBC and RBC in *Trachinotus ovatus* [49], *Rutilus Frisii kutum* [19], and *Labeo rohita* [50]. It seems that the response of hematological indicators in fish fed β-glucan supplements depends on the condition of environmental stress.

Inflammatory cytokines exhibited a pattern of different gene expression in the spleen, intestine, and head kidney. In the spleen, the increase in *TNF-α* expression was significantly decreased by dietary β-glucan supplementation. *TNF-α* is produced primarily by monocytic lineage cells, such as macrophages and *TNF-α*-activated macrophages, which can cause many autoimmune diseases [51]. In the spleen section, macrophage disappearance is reflected by the reduction of gene expression of *TNF-α* in tilapia fed β-glucan in brackish water. In macrophages, as a receptor, *TNF-α* can regulate the production of *TGF-β1* through the c-Jun NH2 terminal kinase pathway. However, this pathway in macrophages acts a negative autoregulatory loop in the control of *TNF-α*-induced *TGF-β1* production [52]. Therefore, gene expression of *TGF-β1* also exhibited a significant decrease in the spleen of tilapia fed β-glucan in brackish water. *TGF-β1* in the intestine showed a different expression pattern compared to that in the spleen. *IL-1β* is a pro-inflammatory cytokine that can initiate and regulate the inflammatory process, as well as apoptosis and cell division processes, and this gene is important for stimulating effective innate host defenses, cell multiplication, phagocytosis, and nitric oxide production [53,54]. Supplementation with β-glucan can significantly suppress the expression of *IL-1β* in the spleen, intestine, and head kidney of tilapia. Similarly, β-glucan supplementation can also down-regulate the expression of *IL-1β* in the head kidney and intestine of *Cyprinus carpio* L [55]. The *MIF* has gained substantial attention as a pivotal upstream mediator of innate and adaptive immune responses and can serve as a biomarker for diseases with inflammatory reaction [56]. The gene expression of *MIF* in the intestine and spleen was decreased due to β-glucan intake in brackish water. β-glucan has a dual function of both pro-inflammatory mediator production and anti-inflammatory properties [57,58]. In the present study, the enhanced immune response in the spleen induced by environmental salinity could be significantly relieved by dietary β-glucan. However, pro-inflammatory indicators, such as *TNF-α* in the intestine and head kidney, were still significantly increased by the supplementation with β-glucan in brackish water.

The health status response of the intestine was also reflected in the results of the microbial community. β-glucan can be absorbed by the intestine and stimulate immune response [59]. It can be used as a probiotic fermentation substrate which is catabolized by anaerobic bacteria to produce SCFA, modulate the production of cytokines and chemokines, and improve systemic inflammation [60]. SCFA is also a vital nutrient substrate for mucosa and epithelial cells, improving intestinal epithelial cell proliferation and promoting peristalsis [14,15]. Intestinal microbiota richness and diversity estimators were decreased due to the supplementation of β-glucan, especially in tilapia fed 0.4% β-glucan. In *Solea senegalensis*, three different glucan diets all reduced microbiome species richness and diversity by oral intubation [61]. The reduction of richness and diversity was also reported in *Apostichopus japonicus* and *C. carpio* L fed β-glucan [20,62]. Similar results were found in *L. vannamei* fed β-glucan under salinity stress [12]. Although bacterial diversity and richness can play essential roles in maintaining intestinal ecological function, the vital effect still depends on the enrichment of certain beneficial species in the intestinal microbiota community, rather than diversity itself [63,64]. Additionally, PLS-DA indicated that β-glucan supplementation can modulate the intestinal microbiota community of tilapia, and the modulated community was similar to that in the freshwater control, especially in tilapia fed 0.4% β-glucan. This is indicative that β-glucan supplementation in the diet can restore and benefit the intestinal microbiota environment in tilapia in brackish water.

With respect to the constituents of the intestinal microbiota at the genus level, 0.2% and 0.4% β-glucan supplementation significantly increased the abundance of *Lactobacillus* in tilapia. Lactic acid bacteria are beneficial to the fish intestinal ecosystem due to producing bacteriocins, lactic acid, and other antagonistic compounds to inhibit the amounts of some fish pathogens [65]. Previous research indicated that β-glucan is a fermentative substrate that enhances the proliferation of *Lactobacillus* in vitro [66]. *Lactobacillus plantarum* is an effective additive in the tilapia diet [67]. The dose of 0.4% β-glucan in the diet significantly increased the abundance of *Nitrospiraceae* in the intestine of tilapia as compared with the abundance observed in fish in brackish water fed the basal diet. This may present a close relationship with a low pH in the intestine due to high richness of *Lactobacillus* and *Nitrospiraceae,* which are more commonly found at a pH < 2.0 [68]. *Phycicoccus* and *Collinsella* also showed significantly-higher abundance in tilapia fed β-glucan. These two groups of microorganisms showed higher abundance in the intestine of *Eriocheir sinensis* and canines when fed diets with L-tryptophan and green tea polyphenols, respectively [69,70]. Compared to tilapia in brackish water fed a basal diet, 0.4% β-glucan in the diet significantly increased the richness of *Streptomyces* in the intestine. *Streptomyces* fungus can produce chitinase in the intestine to inhibit pathogens [71]. *Streptomyces cerevisiae* β-glucan can also prevent airway hyperreactivity and pulmonary inflammation in a murine asthma model [72]. At the family level, 0.2% β-glucan supplementation significantly decreased the abundance of *Thermoactinomycetacae*, *Caldilineaceae*, *Solirubrobacteraceae*, *Microbacteriaceae* and *Phyllobacteriaceae* in the intestine of tilapia as compared with those in the freshwater group. The *Caldilineaceae* abundance can be inhibited by resveratrol and oxytetracycline intake in tilapia and zebrafish, respectively [73,74,75]. Meanwhile, resveratrol can also increase the abundance of *Firmicutes,* just like β-glucan in tilapia [74]. Compared to the results without β-glucan supplementation in both freshwater and brackish water, β-glucan supplementation in the diet could increase the abundance of *Rikenellaceae* in the intestine of tilapia. The richness of *Rikenellaceae* is positively correlated with the inflammatory foci and the alleviation of intestinal barrier dysfunction [76]. At the phylum level, 0.4% β-glucan significantly increased *Firmicutes* abundance in tilapia intestine compared with fish in brackish water fed a basal diet. Similar results were found in *L. vannamei* and *Acipenser baerii* fed β-glucan and arabinoxylan oligosaccharides, respectively [12,77].

The intestinal microbial communities of *O. niloticus* fed with 0.4% β-glucan showed the highest average connectivity and most complex ecological networks among all the treatments. Interacting intestinal bacteria can achieve a dynamic balance in an organism, though they are still sensitive to environmental and nutritional factors [78]. In addition to the characteristics of ecological network complexity, intestinal microbial stability is also closely related to the forms of interspecific interaction with similar or complementary functions [79]. Positive interactions indicate complementation or cooperation, while negative interactions may show competition or predation between bacteria at the family level [80]. Supplementation with 0.2% β-glucan can significantly increase the ratio of positive interaction, and the 0.4% β-glucan supplementation can significantly increase the negative interactions of the intestinal microbiota in *O. niloticus*. Positive interactions can improve evolutionary stability when families are more likely to interact with other cooperators [81]. However, the intestinal microbiota may be more stable when the community is more competitive than cooperative. Therefore, 0.4% β-glucan supplementation can provide a more stable intestinal microbial community in *O. niloticus* than 0.2% β-glucan supplementation and the basic diet.

## 5. Conclusions

Dietary supplementation with 0.2% and 0.4% β-glucan can significantly relieve the spleen enlargement and improve the immune function of *O. niloticus* in brackish water. Meanwhile, β-glucan intake (especially 0.4% concentration) also can optimize the intestinal microbiota, enhance the complex microbiota interactions of *O. niloticus* in brackish water and improve fish health.

## Figures and Tables

**Figure 1 animals-10-02243-f001:**
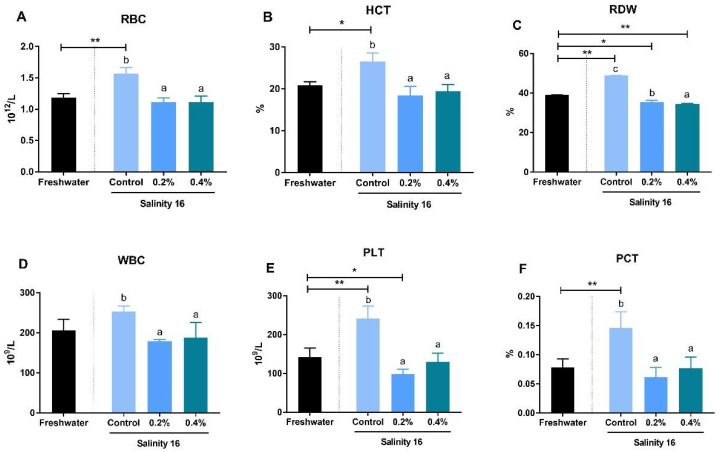
Effects of β-glucan supplementation on hematology in *O. niloticus* in freshwater and brackish water. Values are represented as mean ± SE. A single asterisk (*) indicates a significant difference (*p* < 0.05), and two asterisks (**) indicate an extremely-significant difference (*p* < 0.01). (**A**) red blood cell count (RBC); (**B**) hematocrit (HCT); (**C**) red cell distribution width (RDW); (**D**) white blood cell count (WBC); (**E**) platelet count (PLT); and (**F**) plateletcrit (PCT). Different letters with a–c indicated significant differences among three treatments in 16 salinity water (*p* < 0.05).

**Figure 2 animals-10-02243-f002:**
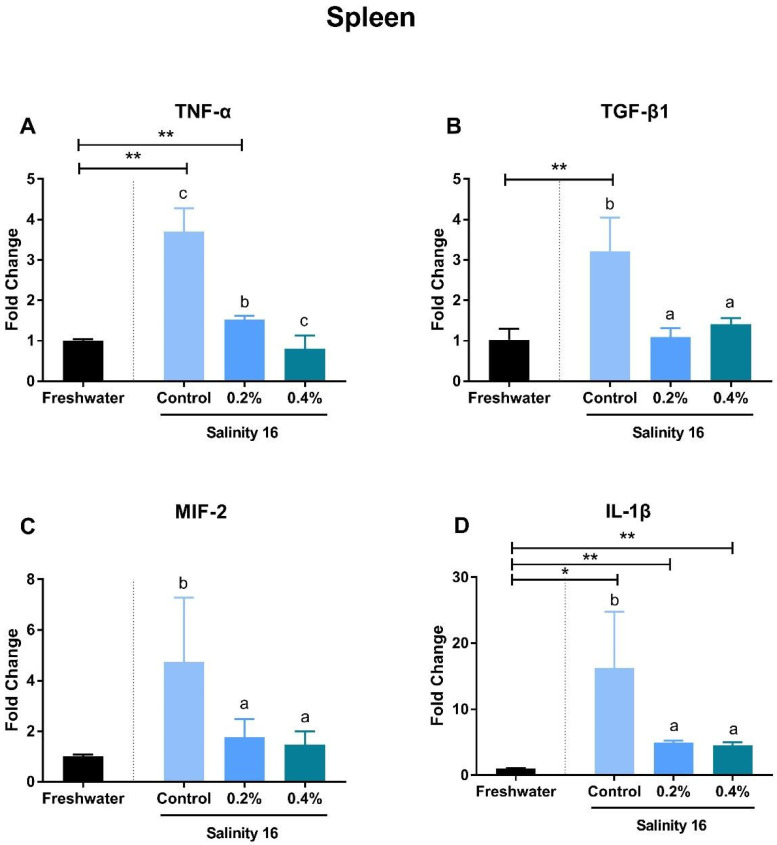
The gene expression of *TNF-α* (**A**), *TGF-β1* (**B**), *MIF-2* (**C**), and *IL-1β* (**D**) in the spleen of *O. niloticus* fed diets with β-glucan in freshwater and brackish water. A single asterisk (*) indicates a significant difference (*p* < 0.05) and two asterisks (**) indicate an extremely-significant difference (*p* < 0.01). Different letters with a–c indicated significant differences among three treatments in 16 salinity water (*p* < 0.05).

**Figure 3 animals-10-02243-f003:**
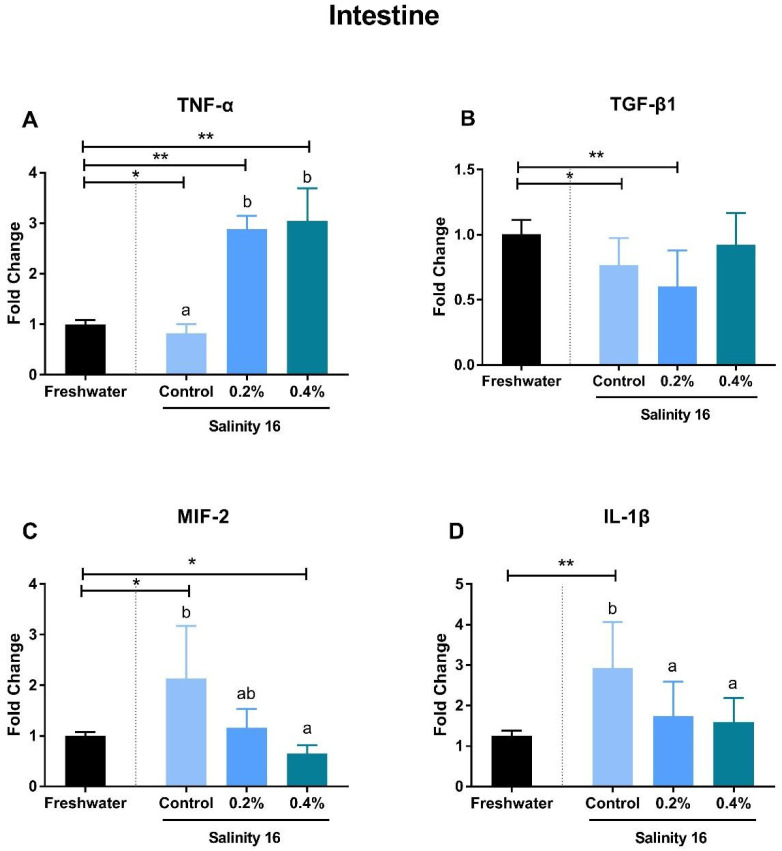
The gene expression of *TNF-α* (**A**), *TGF-β1* (**B**), *MIF-2* (**C**), and *IL-1β* (**D**) in the intestine of *O. niloticus* fed diets with β-glucan in freshwater and brackish water. A single asterisk (*) indicates a significant difference (*p* < 0.05), and two asterisks (**) indicate an extremely significant difference (*p* < 0.01). Different letters with a,b indicated significant differences among three treatments in 16 salinity water (*p* < 0.05).

**Figure 4 animals-10-02243-f004:**
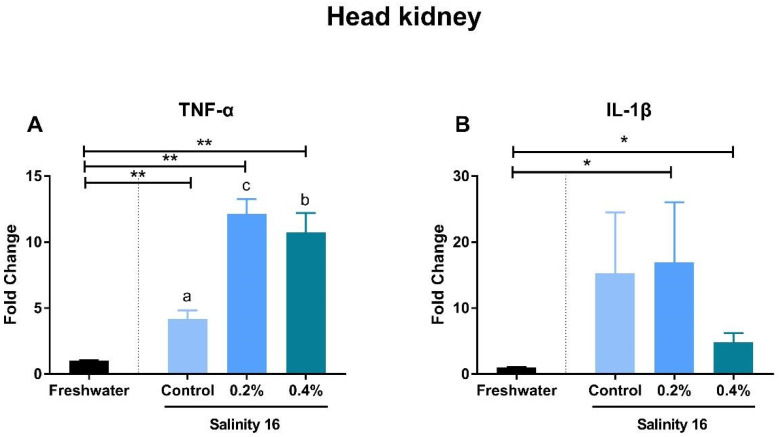
The gene expression of *TNF-α* (**A**) and *IL-1β* (**B**) in the head kidney of *O. niloticus* fed diets with β-glucan in freshwater and brackish water. A single asterisk (*) indicates a significant difference (*p* < 0.05), and two asterisks (**) indicate an extremely-significant difference (*p* < 0.01). Different letters with a–c indicated significant differences among three treatments in 16 salinity water (*p* < 0.05).

**Figure 5 animals-10-02243-f005:**
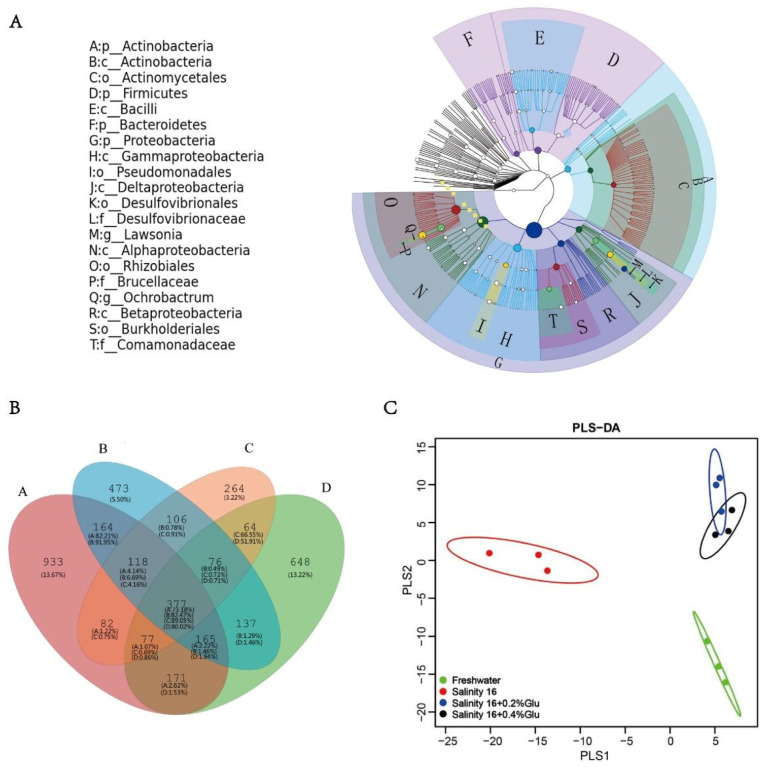
Cladogram (top 20), Venn diagram, and partial least squares discrimination analysis (PLS-DA) diagram of the intestinal microbiota in *O. niloticus* fed diets with β-glucan in freshwater and brackish water. (**A**) Cladogram of the intestinal microbiota displaying the top 20 components as inferred by GraPhlAn. Node size is proportional to the average abundance. Color indicates the relative concentration of the clusters. (**B**) Venn diagram showing the distribution of all OTUs shared and unique in *O. niloticus* reared in different environmental salinity conditions and fed diets with or without β-glucan (0, 0.2%, and 0.4%). The percentage indicates the ratio of correlated OTUs in the total sequences of each treatment. A: 16 psu, B: 16 psu + 0.2% β-glucan, C: 16 psu + 0.4% β-glucan, D: freshwater. (**C**) PLS-DA discriminant analysis of intestinal microorganisms in *O. niloticus* under different environmental salinity conditions and fed diets with or without β-glucan (0, 0.2%, and 0.4%). Each dot represents a sample, the points with the same color belong to the same group, and the marked points of groups are marked with ellipses. If the samples belonging to the same treatment are closer to each other, and the distance between the different treatment points is farther, the classification model will be more reliable.

**Figure 6 animals-10-02243-f006:**
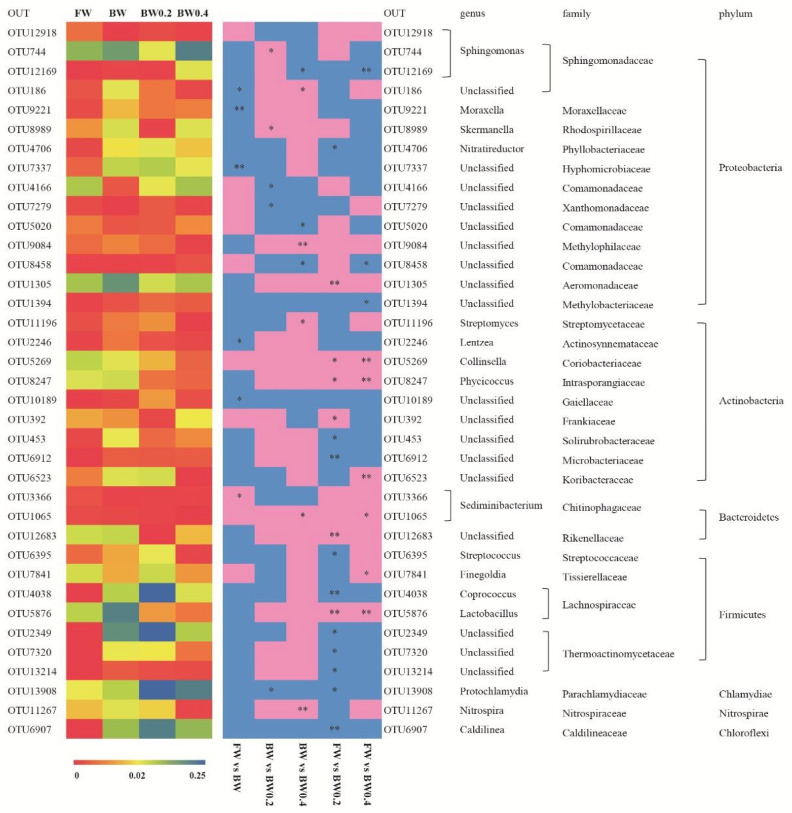
Heatmap analysis of 37 OTUs. The left heatmap includes the color bar of each OTU average value of three samples in each treatment. The right heatmap includes the color bars representing the difference of OTU in *O. niloticus* between two treatments. The pink bar suggests that the abundance of OTUs is higher and the blue bar suggests that the abundance of OTUs is lower. A single asterisk (*) indicates a significant difference (*p* < 0.05), and two asterisks (**) indicate an extremely-significant difference (*p* < 0.01). The taxonomy of the OTUs (genus, family, and phylum) is depicted on the figure’s right. FW denotes freshwater; BW denotes brackish water; BW 0.2 and BW 0.4 denote brackish water with 0.2% and 0.4% β-glucan, respectively.

**Figure 7 animals-10-02243-f007:**
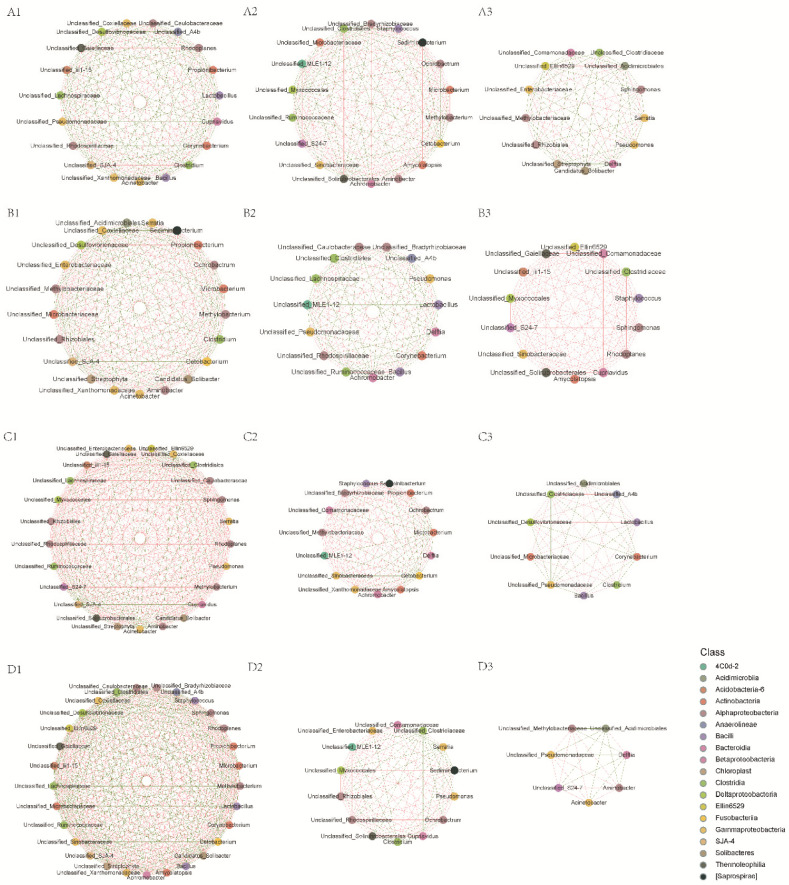
Interspecies interaction network of the intestinal microbiota for *O. niloticus* among four treatments. Each node represents a bacterial OTU. Node colors indicate OTUs affiliated with different major classes. The green edge indicates negative interaction, and the red edge indicates positive interaction between two individual nodes. (**A1**,**A2**,**A3**): freshwater; (**B1**,**B2**,**B3**): 16 psu; (**C1**,**C2**,**C3**): 16 psu + 0.2% β-glucan; and (**D1**,**D2**,**D3**): 16 psu + 0.4% β-glucan.

**Table 1 animals-10-02243-t001:** Ingredient formulation (g/kg dry basis) of three experimental diets fed to *Oreochromis niloticus.*

Ingredients	β-glucan Contents (%)		
	0	0.20	0.40
Fish meal	80.00	80.00	80.00
Soybean meal	450.00	450.00	450.00
Maize meal	250.00	250.00	250.00
Wheat middlings	74.00	74.00	74.00
Maize oil	36.00	36.00	36.00
Dicalcium	10.00	10.00	10.00
Vitamin premix ^a^	5.00	5.00	5.00
Mineral premix ^b^	5.00	5.00	5.00
Cellulose	60.00	57.73	55.46
Carboxymethyl cellulose	30.00	30.00	30.00
β-glucan ^c^	0.00	2.27	4.54
Total	1000.00	1000.00	1000.00

^a^ Vitamin premix (mg/kg): vitamin A (500,000 IU/g), 8 mg; vitamin D3 (1,000,000 IU/g), 2 mg; vitamin K, 10 mg; vitamin E, 200 mg; thiamine, 10 mg; riboflavin, 12 mg; pyridoxine, 10 mg; calcium pantothenate, 32 mg; nicotinic acid, 80 mg; folic acid, 2 mg; vitamin B12, 0.01 mg; biotin, 0.2 mg; choline chloride, 400 mg; vitamin C-2- polyphosphate (150 mg/g vitamin C activity), 400 mg. ^b^ Mineral premix (mg/kg): zinc (ZnSO4·7H2O), 50.0 mg; iron (FeSO4·7H2O), 40 mg; manganese (MnSO4·7H2O), 15.3 mg; copper (CuCl2), 3.8 mg; iodine (KI), 5 mg; cobalt (CoCl2·6H2O), 0.05 mg; selenium (Na2SeO3), 0.09 mg. ^c^ β-glucan in this study was extracted from yeast cell walls and purchased from Swiss ABAC. Research and Development Co., Ltd (Barden, Swiss).

**Table 2 animals-10-02243-t002:** Primers designed in the qPCR.

Gene	5′-3′ Primer Sequence	GenBank Accession No.
*IL-1β*	F: GAGCACAGAATTCCAGGATGAAAG	XM_019365841.1
	R: TGAACTGAGGTGGTCCAGCTGT	
*MIF-2*	F: AGCAGAAGCAGGAAGGCGAAGA	XM_003444573.4
	R: CGGTACATCACCTCTGGCAACATT	
*TGF-β1*	F: AAGAGGAGGAGGAATACTTTGCCA	XM_003459454
	R: GAAGCTCATTGAGATGACTTTGGG	
*TNF-α*	F: CAGAAGCACTAAAGGCGAAGAACA	NM_001279533
	R: TTCTAGATGGATGGCTGCCTTG	
*EF-1α*	F: ATCAAGAAGATCGGCTACAACCCT	XM_005469373.3
	R: ATCCCTTGAACCAGCTCATCTTGT	

*IL-1β*, interleukin-1β; *MIF-2*, migration inhibitory factor-2; *TGF-β1*, transforming growth factor-β1; *TNF-α*, tumor necrosis factor-α; *EF-1α*, elongation factor-1α.

**Table 3 animals-10-02243-t003:** Growth performance, survival, and relative spleen weight of *Oreochromis niloticus* fed diets with different β-glucan percentages under two salinities for 8 weeks.

Items	Freshwater	Brackish Water (16 psu)-β-glucan (%)		
		0	0.2	0.4
Initial weight	1.27 ± 0.01	1.28 ± 0.01	1.28 ± 0.01	1.27 ± 0.01
Final weight	38.03 ± 2.86	41.50 ± 1.53	38.67 ± 2.29	38.40 ± 2.70
Weight gain (%)	2974.75 ± 226.02	3235.35 ± 109.63	3025.67 ± 172.76	3008.85 ± 206.30
Relative spleen weight (%)	0.29 ± 0.01 ^b^	0.37 ± 0.03 ^c^	0.24 ± 0.09 ^a^	0.23 ± 0.02 ^a^
Survival (%)	100	100	100	100

Values of initial weight and final weight are means of three measurements. Values of relative spleen weight are means of 15. Different superscript letters indicate significant differences within the same row (c indicated the highest value) (*p* < 0.05).

**Table 4 animals-10-02243-t004:** Summary of diversity index (Shannon and Simpson) and estimated OTU richness (Chao 1 and ACE) for intestinal bacterial diversity analysis of *Oreochromis niloticus* fed diets with different β-glucan percentages under two salinities for 8 weeks.

Items	Freshwater	Brackish Water (16 psu)-β-glucan (%)		
		0	0.2	0.4
*Richness estimate*				
Chao 1	782.01 ± 138.12 ^ab^	939.67 ± 63.50 ^b^	758.00 ± 125.74 ^ab^	564.00 ± 16.07 ^a^
ACE	782.30 ± 138.15 ^ab^	939.67 ± 63.57 ^b^	758.11 ± 16.06 ^ab^	564.11 ± 16.06 ^a^
*Diversity estimators*				
Shannon	6.10 ± 0.13 ^ab^	7.02 ± 0.40 ^b^	6.01 ± 0.71 ^ab^	5.46 ± 0.11 ^a^
Simpson	0.94 ± 0.01 ^ab^	0.97 ± 0.01 ^b^	0.94 ± 0.02 ^ab^	0.93 ± 0.01 ^a^

All data are represented as mean ± S.E. (n = 3). Different superscript letters indicate significant differences within the same row (b indicates the higher value) (*p* < 0.05).

**Table 5 animals-10-02243-t005:** Topological properties of the intestinal microbiota co-occurrence network.

Items	Freshwater	16 psu	16 psu + 0.2% β-glucan	16 psu + 0.4% β-glucan
Node	50	50	50	50
Edge	402	406	450	518
Average degree	16.08	16.24	18	20.72
Diameter	1	1	1	1
Graph density	0.328	0.331	0.367	0.423
Centralization	0.041	0.080	0.128	0.155
Heterogeneity	0.153	0.198	0.347	0.426
Average clustering Coefficient	1	1	1	1
Average path length	1	1	1	1
Positive/negative association (%)	53.23/46.77	64.53/35.47	76/24	55.02/44.98

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
