# Peer review of "Recovery from Hypersaline-Stress-Induced Immunity Damage and Intestinal-Microbiota Changes through Dietary β-glucan Supplementation in Nile tilapia (Oreochromis niloticus)"

_animals, 2020, doi:10.3390/ani10122243_

Round 1

Reviewer 1 Report

All minor revisions have been satisfied

Author Response

Thank you for your approval and hard work.

Reviewer 2 Report

The original version has been substantially modified according to reviewer suggestions, therefore I think it is suitable for publication. Please note that the current version has marks from Word's changes tracking, hence the authors should upload the final version after changes, without marks.

I would like to recommend authors to include this article in the current topic "New Approaches to Fish Welfare", available in Animals.

Author Response

Thank you for your comments. We have upload the final version without marks. If we have a suitable manuscript, we will choose the topic “New Approaches to Fish Welfare”. Thank you for your advice.

Reviewer 3 Report

I think it's a interesting topic for animal welfare nowadays, regarding one of the most farmed fresh water fish species.

However, I believe that this manuscript not need big changes as suggested

I think you can publish the manuscript after major revision.

Author Response

Thank you for your comments. We have checked the manuscript carefully. Thank you for your suggestion.

This manuscript is a resubmission of an earlier submission. The following is a list of the peer review reports and author responses from that submission.

Round 1

Reviewer 1 Report

The MS offers interesting cues on the immunomodulatory role dietary of β-glucan in tilapia under hypersaline stress and its effect on microbiota community. However, even if the immune bosting effect of this additive has been widely accepted, the present study lacks of several evidences showing beneficial effects at least on gut welfare.

Overall, the methodological approach is often inadequate and results description often merely speculative. In addition, the authors lack of basilar histological and anatomical knowledge,  Also, the molecular section must be improved.

On the contrary, microbiome analysis is accurate and well described both in the methods and in the results, so that it seems that has been performed and styled by separate research group and writers. However, how hypersaline challenges may affect tilapia bacterial community is it is not very thoroughly discussed.

In particular, just some comments:

-  the experimental design lacks of two important groups including freshwater+ β-glucan (at the two % used)

- It is better to refer to “mucosal/intestinal folds”, since the term “villi” (even if used) is inappropriate when dealing with fish, for a set of histological implication, whit respect to birds and mammals

- No attempt to quantify histological changes has been made (https://doi.org/ 10.3390/ani10020231; https://doi.org/10.1016/j.fsi.2008.02.013.), so that results are merely descriptive and speculative

- 2.7. (line 138) – this section should be referred as “histological analysis” (not “paraffine section)

- (line 141) – what  “vacuum-embedding” is meant by?

- 2.7. (line 145) – mRNA quantification is not mRNA expression (which doesn’t exists). You should refer to “gene expression”  

- 3.1 (line 202) “Survival, growth performances and relative spleen weight” instead of “Growth performances”. These results should be better explained

- (line 211) “Histologycal results” instead of “Paraffine sections….”. This section is very reductive, and should be deeply investigated in a study on pre-biotics, as in this case

- 3.4 (line 219) “GENE EXPRESSION”. Results should be deeply described and a possible dose-dependent effect should be emphasized

- Fig. 1. (d) – there are no significant differences between freshwater groups and the group you named “Control”?

- Fig. 3. (b) – probably an artefact, since no inflammatory event is appreciable. These illustrations do not show any appreciable histological alteration

- Fig. 3 (caption and letters in the fig.) – please refer as folds. I cannot understand why the terms “villi” and “folds” are separately used.

-  Fig. 4. (c) – there are no significant differences between freshwater groups and the group you named “Control”?

- Intestine cytokines gene expression, as presented, does not show β-glucan beneficial effect, at least in terms of tnf-a and tgf-b1 mRNA abundance. How do the authors explain and discuss this result?

- Kidney cytokines gene expression is controversial. How do the authors explain the tnf- and il-1b results?

- (line 417-418) – the authors cannot assert that “the MMCs disappeared” without any attempt of quantification, similarly to intestine histological results

-  (line 215) – where the authors demonstrate that the goblet cells were broken? Overall, what does it means? Goblet cells quantification should be performed.

-  (line 497) – these interesting result have been obtained in fish fed insects (rich in chitin, a β-glucan analogous) DOI: 10.1016/j.aquaculture.2019.734659

- Overall language and writing style have to be accurately and deeply improved

Author Response

  1. Positive function of β-glucan have been confirmed in various research. In our study, the focus was to detect the recovery effect of β-glucan on the negative effects of tilapia in brackish water environment. Therefore, we did not set these two treatment with β-glucan in freshwater.
  2. Yes, we agree. Intestinal fold is more appropriate for fish. We have changed the expression to “intestinal fold”. (Line 36, 145, 219, 300, 302, 464 and 465)
  3. The results of the histology are obvious, no statistical analysis in previous manuscript.

According to your comments, we have analyzed intestinal fold height, goblet cell number and melano-macrophage center area of tilapia under experimental treatments and added statistical data behind the histological pictures. (Figure 2-E and Figure 3-E, F)

  1. “Paraffine section” has been changed to “Histological analysis”. (Line 138)
  2. “Vacuum-embedding” means waxing and embedding. “Vacuum” means there was no bubbles in paraffin.
  3. Description of “mRNA expression” were all changed to “gene expression” through the manuscript.
  4. The title has been changed to “Survival, growth performance and relative spleen weight”. (Line 205)
  5. “Paraffine sections” has been changed to “Histological results”. (Line 214)

As a critical tissue for immune barrier and nutrient absorption, intestine histology response significantly after intake of prebiotics. It is a very valuable research topic indeed and we will carry out further research in the future.

  1. We did not discuss the content of dose-dependent result due to less concentration gradient of β-glucan in diets. There are only two typical supplement concentration in diets for teleost fish in this study.

According to your comment, we have added appropriate discussion in line 225-226.

  1. Yes, there was no significant difference between these two treatments by the t-test.
  2. All histological pictures were real through analysis and screening of several individuals. There was obvious damage of intestinal fold in Figure 3-B. We have added statistical data behind the histological pictures.
  3. Thank you for your comment. We have modified the annotation in pictures. (Figure 3 and Line 302)
  4. Yes, there was no significant difference between these two treatments by the t-test.
  5. In the results of gene expression, TNF-α showed significantly higher gene expression in intestine of tilapia fed diets with β-glucan than that in control. The experimental period was eight weeks, during which the β-glucan diets were continuously fed without interruption. Immunosuppression may occur after long-term immune activation in different tissues of organism. Furthermore, different tissues show distinguishing sensitive to dietary immunomodulator which may induce very different results.
  6. Tissue and gene can show different sensitive to immunomodulator supplement with different concentration and time in organism. Long-term and high dose intake may induce immunosuppression or immune exhaustion which reflected in decreased gene expression of IL-1β in tilapia fed diet with 0.4% β-glucan than that fed 0.2% β-glucan. Not all cytokines are upregulated, especially after long-term immune stimulation.
  7. Thank you, we have added quantification analysis behind histological presentation. (Line 288)
  8. Quantitative data were added in line 296. In teleost, intestinal goblet cells reflect both the histological integrity and also the digestive enzymes secretion of organism. Significantly decrease goblet cells number in tilapia under hypersaline water ambient showed unnormal histology of intestine compared to tilapia in freshwater.

In figure 3-B, the top of the intestinal fold has obvious structural damage which also affect the goblet cells at the top of the intestinal folds.

  1. The reference has been added in line 502 and 762.
  2. The language and writing style have been accurately and deeply improved by professional English editor.

Reviewer 2 Report

Revision

Regarding manuscript revision (ID animals-952605), the effect of dietary β-glucan supplementation in tilapia farmed in brackish water was investigated. Authors demonstrated that β-glucan supplementation could significantly decrease hypersaline-stress induced immunity damage in Nile tilapia (Oreochromis niloticus) spleen and intestine. Moreover, this substance added to the feed, showed beneficial effects on the immune system and significantly improves the quality of the intestinal microbiota. In the believe this study presents innovative research result, I would suggest some minor revisions.

Mat and met

Paragraph 2.2

  • Line 107: Why all male fish? Moreover, at 1,28 g tilapia are in larval stage, so fish do not exhibit sexual differentiation. Fingerlings have to be grown until it is possible to distinguish sex, and became male by feeding with masculinising hormone, temperature and genetic manipulation. Fuentes-Silva, Carlos et al. “Male tilapia production techniques: A mini-review.” African Journal of Biotechnology12 (2013): Please clarify

  • Line 121: you should insert some references for pain release procedures. I would suggest the following:
  • Readman, G. D., Owen, S. F., Knowles, T. G., & Murrell, J. C. (2017). Species specific anaesthetics for fish anaesthesia and euthanasia. Scientific reports, 7(1), 1-7.
  • Iaria, C., Saoca, C., Guerrera, M. C., Ciulli, S., Brundo, M. V., Piccione, G., & Lanteri, G. (2019). Occurrence of diseases in fish used for experimental research. Laboratory animals, 53(6), 619-629.

  • Line 295 add “;” damaged after intestinal villi

  • Have other tissues been examined such as gills? If so, were there histological differences between the control and experimental groups? Beta glucan had a protective effect in this tissue? Example, mucus production, etc?

Discussion

Line 413: teleost does not have lymphatic gland, please correct with lymphoid organ.

Author Response

  1. Yes. Tilapia was treatment by masculinising hormone stimulation to convert sex to male at 10 mm in lenght. In terms of production, after years of certification, the conversion rate of males can reach 99.5 percent. This is also the experimental fish that we used in the trial.

At sampling, no gonadal development was observed which is often found when both genders exist simultaneously. Therefore, we call it all male in manuscript.

  1. Thank you for your comment. These two reference have been added in line 122. Reference 35 and 36 have added in line 642 to 646.
  2. “;” was added in line 302.
  3. In this trial, gill histology was not be examined. This is an enlightening question, we will detected this change in the further research. Besides, we found a relevant paper can be reference.

Mahmoud, A.O.D.; Safaa, E.A.; Mahmoud, S.G.; Eman, M.M.; Moustafa, S.S.; Marwa, F.A.; Awatef, H.H.; Amira, A.O.; Rasha A.A. The influence of dietary β-glucan on immune, transcriptomic, inflammatory and histopathology disorders caused by deltamethrin toxicity in Nile tilapia (Oreochromis niloticus). Fish and Shellfish Immunology. 2020, 98, 301-311. DOI: https://doi.org/10.1016/j.fsi.2020.01.035.

  1. Thank you. “lymphatic gland” have been changed to “lymphoid organ” (Line 419).

Round 2

Reviewer 1 Report

Even if some improvements to the text have been applied, serious flaws are still present, making the scientific solidity of the MS faltering. 

Asserting "The results of the histology are obvious" is the least scientifically correct sentence that can be reported in a review response, since nothing is "obvious" in absence of quantitative results. In addition, I would like to further underline that histological pictures do not allow to appreciate any inflammatory events, including leucocytes infiltrate, widening of submucosa or the mucous cells reduction described in the results. The magnification is too low to appreciate any of the above mentioned histological alterations. The ephytelial exfoliation showed in fig. 3b is not explicative, alone, unless the authors do not use this parameters and consider it as a statistically significative occourence on a significant number of fish and sections analysed. The number of sections and optical fields used for histological analysis is not mentioned, thus leading to the conclusion that the authors stopped to mere qualitative description. Moreover, there isn't any metodologycal description of the criteria adopted to perform histological quantitative/semiquantitative analysis. The number of the sections analyzed, how the measurements were performed, how the mucous cells count has been performed, etc, is not specified in the M&M section, while the existing letterature is now plenty of accurate metodological descriptions of histological/morphometric analysis. In addition, moving to the added results, I have serious perplexities. How it is possible that the maximum folds heigh detected (fig.3e) is about 60 micron? This is morphologically impossible and such measurements, cannot be plausible. Which is the number of sections/area/folds used for mucous cells measurements? I suggest to refer to an histology expert in order to provide solid evidences to the MS; otherwise you should remove histology from this paper.    

The language must be further improved:

  • line 27: ...whas given the high expectation...
  • line 34: ...a significantly more reduction...
  • line 53: ...can enhance by...
  • line 60: ..except for.. -->beside...
  • line 65: ...supplement --> supplementation
  • line 91: health status's...
  • line 116: ...animals were lighted by...
  • line 143-144: The thikness of the slice is...
  • and so on..

Author Response

We have really gained a lot from your comments. Thank you. The language have been further improved by professional English editor. Details of the revision are shown in the updated manuscript.

Thank you for your comments. We have also referred to several papers about teleost histology before writing this manuscript. “Obvious” is a really bad description and we have modified the statement. We will use scientific descriptions in later writing.

In the section of results, we only described the change of intestinal folds height, goblet cell number and epithelial cells exfoliation. There was no inflammatory events in histological results. Inflammatory events were analyzed in the section of gene expressions of immune-related genes in the spleen, head kidney and intestine of tilapia.

Epithelial cell exfoliation was a significative occurrence event in intestine of tilapia in brackish water environment. In the four intestine samples we analyzed, epithelial cells were exfoliated significantly in three fish. This result was added in line 234 and 322.

Image J was used to analysis the intestinal fold height, goblet cell number and melano-macrophage center area in tilapia. Methodological description was added in line 151 to 159. Histology expert who are experienced in histology give a lot of advice and guidance.

About the results of intestinal fold height, we have put a wrong scale. So, we used new software and methods to re-detect the height of intestinal folds. Correct results were showed in line 316 (Figure 3-E and F).
